# A Study on Single Pilot Resource Management Using Integral Fuzzy Analytical Hierarchy Process

Kwang Hyun Im [1], Woongyi Kim [1] and Seock-Jin Hong [2],*

1 Department of Air Transportation and Logistics, Hanseo University, Taean-gun 32158, Korea; jaralim4@gmail.com (K.H.I.); wykim@hanseo.ac.kr (W.K.)
2 Department of Logistics and Operations Management, G. Brint Ryan College of Business, University of North Texas, Denton, TX 76203-5017, USA
* Correspondence: seock.hong@unt.edu; Tel.: +1-940-369-8911

**Abstract:** This research aims to help develop aviation safety policies for the general aviation industry, especially for flight training schools. The analytical hierarchy process (AHP), fuzzy AHP, and fuzzy integral methods were used to find variables that impact aviation safety for training pilots in Korea and the United States using survey participants' experience and perceptions. The results represent the circumstances of aviation safety in the real world where single pilot resource management, especially situational awareness, is crucial. The authors find that integral fuzzy AHP provides more explicit considerations, making up for the ambiguity of the linguistic responses caused by the AHP and fuzzy AHP.

**Keywords:** aviation safety; pilot behavior; situational awareness; crew resource management; AHP; fuzzy AHP

## 1. Introduction

Pilots who work in complex environments are routinely exposed to high amounts of situational stress in the workplace, inducing pilot error, which may result in a threat to flight safety [1]. This phenomenon places the more significant risk on the flight crew and passengers of an airplane because it increases the chance of pilot mishaps [2]. Fatigue is pervasive among pilots because of irregular working hours, long-haul flights, circadian disruption, and insufficient sleep [3]. Various reasons and factors are indirectly connected to aviation accidents and incidents that seriously threaten aviation safety [4]. Research on the human error framework, human factors analysis and classification systems (HFACS) [5], the SHELL model—software, hardware, environment, liveware, and central liveware [3,6,7]—crew resource management (CRM) [8], and other areas has been developed to reduce and eliminate aviation accidents and incidents caused by human errors.

These human errors have been emphasized only in multi-crew environments under the concept of CRM [9]. While CRM is aimed at pilots operating in multi-pilot environments, the CRM concepts have been utilized for single pilot resource management (SRM) [10,11]. SRM focuses on a single pilot's operations, eliminating the emphasis on the role of the team in pilot training [12]. SRM consists of all the resources available to pilots before and during a flight to enhance the safety and efficiency of single pilot operations [11,13]. A structured approach to SRM helps pilots learn to gather information, analyze it, and make sound decisions for safe flying [13]. Pilots, dispatchers, maintenance personnel, and safety-related personnel should receive CRM/SRM training on an initial and recurrent basis.

However, SRM has not been highlighted as an area of academic research or has focused only on the five Ps (plan, plane, pilot, passengers, and programming). Hence, this current study focuses on the integration of SRM with broader concepts, including not only the five Ps, but aeronautical decision making (ADM), risk management (RM), situational

awareness (SA), automation management (AM), and task management (TM). Decision-making circumstances involve the need to evaluate a finite number of possible choices (alternatives) based on a finite number of attributes (criteria). In selecting a proper aviation safety management system from many alternatives, it is important to remember that those alternatives should consistently provide safety to the people who are continuously working on options for delivering optimal aeronautical safety. The question, therefore, is how to evaluate the alternatives for risk management adequately and to solve hazard issues that continuously occur in the field between people, such as pilots, air traffic controllers, and mechanics, and the aircraft in the air or on the ground. It becomes imperative to build a decision support system for aviation safety or risk that can be visible, direct, convenient, and, of course, interact with the decision makers [14–17].

Making crucial decisions about aviation safety in the aeronautical fields is an everyday activity for all who work with aircraft [10]. Researchers spend lots of time dealing with how and when to act in a decision-making process, which affects the best courses of action before, during, and after a safe flight [18–20]. Choosing the best decision or set of interrelated decisions for reducing and eliminating aviation accidents is an essential task for those who are concerned about flight safety. This current study reviewed a broad range of academic literature. The authors held interviews with student pilots, flight instructors, air traffic controllers, and mechanics who have practical experience in and opinions regarding actual general aviation. The fuzzy AHP of $\lambda$ measure is a special case of fuzzy measures defined iteratively, and is applied to analyze the objective importance of SRM categories and variables. The rest of this paper is organized as follows. Section 2 reviews the related literature for SRM and the categories/variables to support SRM. Section 3 discusses the research methodology. Section 4 presents the methodology and data analysis. Section 5 discusses the results and implementation. Section 6 concludes the study, discusses policy implications, and outlines future research.

## 2. Literature Review

A single pilot operating in general aviation has one of the most demanding civil aviation tasks [21]. Major accidents caused in general aviation are due to poor judgment and decision making and inadequate pre-flight and in-flight planning [13,22,23]. General aviation accident and incident rates far exceed those of the airlines, as do the numbers of people killed or injured and total accidents [22,24–26]. In an attempt to address human error accidents, CRM was developed as a program to train pilot teams in the effective use of non-technical skills [27]. The introduction of a form of CRM training into general aviation could optimize the single pilot's decision-making processes to increase flight safety and improve flight operation efficiency [24]. SRM focuses on single pilot operations, which eliminates the team-oriented training of CRM [28].

Single pilot resource management (SRM) is all about helping pilots learn how to gather information, analyze it, and make decisions [10]. Although the flight is coordinated by a single person and not an onboard flight crew, the use of available resources, such as autopilot and air traffic control (ATC) and automated flight service stations (AFSS), replicates the principles of CRM [20]. The SRM technique involves managing all onboard and outside resources available to a cockpit crew before and during a flight to secure a safe and successful result [10]. Integrating SRM into a general aviation (GA) pilot training program is a vital step toward aviation safety. A structured approach to SRM helps pilots learn to congregate crucial information, examine the information, and make sound decisions during the flight [20].

SRM can be applied using the five-P approach: plan, plane, pilot, passengers, and programming [20]. The plan includes the basic elements of cross-country planning: weather, route, fuel, and current publications, among others [13]. The plan also includes all the events surrounding the flight and allows the pilot to accomplish the mission. The pilot should review and update the plan at regular intervals during the flight, bearing in mind that any of the factors in the original plan can change at any time [29]. The plane includes

the airframe, systems, and equipment, including avionics. The pilot should be proficient in the use of all installed equipment and familiar with the aircraft/equipment's performance characteristics and limitations [30]. As the flight proceeds, the pilot should monitor the aircraft's systems and instruments in order to detect any abnormal indications at the earliest opportunity [29]. A pilot identifies and mitigates physiological errors at all steps of the flight [20].

The passengers can be of considerable help to the pilot by accomplishing tasks, such as those listed earlier. However, passengers can create plausibly dangerous distractions. If the passenger is a pilot, it is also essential to establish who is doing what. The five-P approach reminds the pilot-in-command to consider and account for these factors [20,31]. Programming can refer to both panel-mounted and handheld instruments. The advanced electronic instrument shows how moving map navigators and autopilots can reduce pilot workload and improve pilots' situational awareness [30]. However, the task of programming or operating both installed and handheld equipment (e.g., tablets) can create a serious distraction from other flight duties. This part of the five-P approach reminds the pilot to mitigate this risk by having a thorough understanding of the equipment long before takeoff and by planning when and where the programming for approaches, route changes, and airport information gathering should be accomplished, as well as times it should not be attempted [20,30]. SRM should be used consistently, and solid skills can significantly enhance flight safety [20].

Situational awareness (SA) is the precise perception [4] and understanding of the entire array of resources within the four risk elements that influence safety before, during, and after the flight [20], with internal and external resources found in and out of the aircraft cockpit [10]. All the skills involved in decision making apply to maintaining situational awareness. Keeping up situational awareness requires the use of all flight-related skills and understanding their impact on the safety of flights, as well as using checklists, air traffic controllers, and automated flight service stations [10]. Fatigue, stress, and work overload can reduce the overall situational awareness of the pilot [20].

A literature search for aeronautical decision making (ADM) yields references, definitions, and other relevant information about ADM training in the general aviation environment [10,20]. ADM is a systematic perspective on risk and stress management. Understanding ADM also explains how personal attitudes can impact decision making and helps pilots to adapt those attitudes to improve safety in the flight deck [19]. It is essential to see the factors that cause human beings to make decisions and how the decision-making process works and can be improved [20]. Regardless of the technological developments that enhance flight safety, one important thing remains the same: the human factor, which produces errors [20]. ADM includes three P variables—"perceive" from the given set of conditions for the flight, "process" by evaluating the influence of these conditions on flight safety, and "perform" by acting out the best course of action during a flight [10].

Risk management (RM) includes the PAVE variables—P for the pilot's general health, physical, mental, and emotional state, as well as their proficiency and currency; A for aircraft airworthiness, equipment, and performance capability; V for environment weather hazards, terrain, airports, runways to be used, and conditions; and E for external pressures, such as meetings, people waiting at their destination, et cetera [10]. Pilots perceive hazards by using PAVE to process information and decide whether the identified situation constitutes a risk that should be eliminated, and perform by acting to evaluate the outcome of the hazards [10].

Controlled flight into terrain (CFIT) is when an airworthy aircraft is unintentionally flown into the ground, a mountain, a body of water, or an obstacle under pilot control [32]. In a typical CFIT scenario, the crew is unconscious of the near disaster until too late. CFIT is a significant cause of accidents, causing over 9000 fatalities since the early commercial jet age [33] (Boeing, 2020). Despite the success of advanced technologies, such as the ground proximity warning system (GPWS), enhanced ground proximity warning system (EGPWS), and ground collision avoidance system, at reducing CFIT accidents in the com-

mercial airline industry [32], general aviation aircraft are still not well equipped with this advanced technology.

Automation management (AM) requires a thorough comprehension of how the autopilot system interrelates with the other systems [29]. When flying with advanced avionics, the pilot must know how to control the course deviation indicator (CDI), the navigation source, and the autopilot. Furthermore, a pilot needs to know the peculiarities of the particular automated system being used in the cockpit.

Task management (TM), a significant factor for in-flight safety, is the process by which pilots manage the many concurrent tasks that must be performed to safely and efficiently fly a modern aircraft [20]. A task is a function performed by a human being, as opposed to one performed by a machine (e.g., setting the target heading in the autopilot), and the flight deck is an environment in which potentially many important tasks compete for pilots' attention at any given time [20]. Task management determines which of perhaps many concurrent tasks the pilot(s) attend to at any particular point in time [20]. TM, specifically, requires monitoring continuous tasks to prioritize their status. The prioritization of tasks is established based on their importance, status, and urgency, the allocation of human and machine resources to high-priority tasks, the interruption and subsequent resumption of lower priority tasks, and the termination of completed or no longer relevant tasks [20]. Effective workload management is achieved by planning, prioritizing, and sequencing tasks to avoid work overload [20]. As experience is gained, a pilot recognizes future workload requirements and prepares for high workload periods during periods of low workload [20].

## 3. Discussion

Multiple criteria decision making (MCDM) or multiple criteria decision analysis (MCDA) is a research method for evaluating multiple conflicting criteria in decision making both in daily life and in other settings, such as business, government, and medicine [34]. MCDM and MCDA are also known as collaborative decision making when individuals collectively make a choice from the alternatives before them [35].

MCDM and MCDA can include the analytic hierarchy process (AHP), multi-attribute value theory, and multi-attribute utility theory. Each of these methods has its own characteristics that can be adapted adequately to data analysis. Among these methods, the AHP has more merits than the others when structuring and measuring. The AHP is a structured approach for examining complex decisions [36]. The AHP helps decision makers find the best solutions for their goal of solving the problem [37]. It gives a comprehensive and rational framework for structuring a decision problem, quantifying its elements, relating them to overall goals, and evaluating alternatives [37]. The AHP has a hierarchical decision goal with the alternatives and the criteria for evaluating the alternatives, establishes priorities based on pairwise comparisons of the elements, yields a set of overall priorities for the hierarchy, and checks the judgments' consistency [38].

A hierarchy is a stratified system of ranking and organizing people, things, and ideas, where each element of the system, except for the top one, is subordinate to one or more other elements. Though the concept of a hierarchy is easily grasped intuitively, it can also be described mathematically [39]. Diagrams of hierarchies are often shaped roughly like pyramids, but other than having a single element at the top, there is nothing necessarily pyramid shaped about a hierarchy (see Figure 1). The fundamental nine-point Likert scales of pairwise comparisons are applied [38].

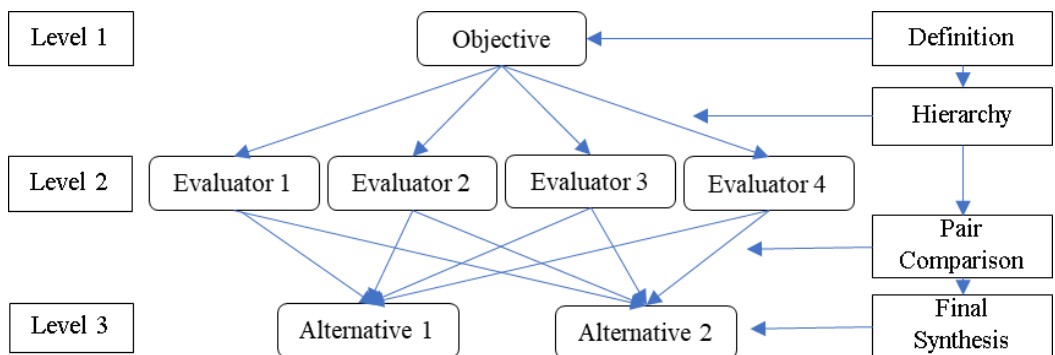

**Figure 1.** Analytical Hierarchy Structure.

Weighting means conducting a pairwise comparison that indicates the relative importance of or preference for evaluation items. As a process of arranging the pairwise comparison values for each problem and calculating the weights for the problem from this, the pairwise comparison matrix $A(a_{ij})$ is drawn up, and the eigenvalue $\lambda_{max}$ of the matrix is calculated. That is, if $\lambda$ involves multiplying an $n \times n$ square matrix $[A]$ by an $n \times 1$ weight matrix $[W]$, the new $n \times 1$ vector matrix $[Y]$ can be acquired, as $[A] \times [W] = [Y]$.

$\lambda_{max}$ is calculated by using components $Y_1 \dots Y_n$ and weights $W_1 \dots W_n$ of $[Y]$, as $(Y_1 W_1 + Y_2 W_2 + Y_3 W_3 + \dots Y_n W_n)/n = \lambda_{max}$.

To survey if there is a logical consistency in the value of the preference index, the consistency ratio (CR) should be tested. The method of calculating a CR is to start from the consistency index (CI). The CI is based on the idea that evaluators would make a consistent judgment in the pairwise comparison, as the eigenvalue $\lambda$ max of the matrix moves closer to the size n of the matrix. CI is defined as CI = $\frac{(\lambda_{max} - n)}{(n-1)}$. Next, the CR is calculated by dividing the CI by the random index (RI), as $CR = \frac{CI}{RI}$, where the RI can calculate the consistency index after drawing up the reciprocal matrix by extracting the integers from one to nine.

The main problems with AHP are the ambiguity and uncertainty arising from the subjectivity of the respondent individual and the problem of inaccuracy caused by the limitations of the expression method; that is, the limit of mathematical theory can distort the results of the AHP and the subsequent decision-making process [40,41]. Therefore, a methodology that can model what systematically gives rise to ambiguity and uncertainty in the decision-making process is required [42,43]. Problems of vagueness and fuzziness have probably always been present in human decision making [44,45]. A fuzzy method is a concept in which the application boundaries vary according to context or conditions instead of being fixed once [46]. The study of fuzzy concepts and language characteristics is called fuzzy semantics [47–49].

Fuzzy AHP is a methodology applied to handle ambiguity and uncertainty effectively [50]. Fuzzy AHP is a systematic approach to an alternative selection and justification problem that uses the concepts of fuzzy set theory and hierarchical structure analysis [51]. It can specify preferences in the form of linguistic or numerical values that are related to the importance of each performance attribute [35]. In the fuzzy AHP method, the pairwise comparisons in the judgment matrix are conducted using fuzzy mathematics and fuzzy aggregation operators [52]. This process enables us to calculate a sequence of weight vectors that can be used to select the main attributes. Decision makers may sometimes not be able to specify preferences between two factors using the nine-point-scaled pairwise comparison [35]. In this current study, we incorporate the traditional AHP to form a "new" fuzzy AHP to address the ambiguous judgments made by the experts during the data collection process [50]. Fuzzy AHP has more advantages than the AHP method, such as deriving pairwise comparison results by using fuzzy numbers, calculating fuzzy triangular numbers by using attributes, and making comparisons between fuzzy triangular numbers and the weights for evaluating group decision-making methods [53].

Let X be a universe of discourse, C be a class of subsets of X, and E, F $\in$ C. A function g: C $\rightarrow$ R where $\varnothing \in \mathbb{C} \Rightarrow g(\varnothing) = 0$ and $E \subseteq F \Rightarrow g(E) \leq g(F)$ is called a fuzzy measure. A fuzzy measure is called normalized or regular if $g(X) = 1$ [54,55].

Fuzzy measures are defined on a semi-ring of sets or a monotone class, which may be as granular as the power set of X, and even in discrete cases, the number of variables can be as large as $2^{|X|}$. A symmetric fuzzy measure is defined uniquely by $|X|$ values. Two important fuzzy measures that can be used are the Sugeno or $\lambda$-fuzzy measure and k-additive measures, introduced by Sugeno [56] and Grabisch [57]. The Sugeno $\lambda$-measure is a particular case of fuzzy measures defined iteratively. Let X = {$x_1, \ldots, x_n$} be a finite set and let $\lambda \in (-1, +\infty)$. A Sugeno $\lambda$-measure is a function of $g$: $2^X \rightarrow [0, 1]$, such that $g(X) = 1$; A, B $\subseteq$X (alternatively, A, B $\in 2^x$) with A $\cap$ B = 0, then $g(A \cup B) = g(A) + g(B) + \lambda g(A) g(B)$. As a convention, the value of g in a singleton set is called a density and is denoted by $g_i = g(\{x_i\})$. Moreover, let X be a finite set, X = {$x_1, \ldots, x_n$} and $g(X)$ be the class of all subsets of X, and thus the fuzzy measure $g(X) = g = \{x_1, \ldots, x_n\}$ can be formulated as $\lambda + 1 = \prod_{i=1}^{n} (1 + \lambda g_i)$ [58,59].

For calculating $\lambda$ with absolute importance for the revision of AHP, we use Formula (1):

$$\prod_{i=1}^{m} (1 + \lambda W_{ij}) - 1 - \lambda = 0 \ (j = 1, \ldots, n) \tag{1}$$

where $i$ is a category and $j$ are variables under each category. The $\lambda$ is less than zero and the relation is $g\lambda(A \cup B) < g\lambda(A) + g\lambda(B)$. As each element interactively includes the influence of others, the individual sum of the influence of each element would be larger than that of each elements' importance. For fuzzy AHP, parameter $C$ is multiplied by the relative importance of AHP; $C$ is the parameter of the fuzzy measure and is attained by the applied boundary condition of Sugeno's $\lambda$ fuzzy measure [56]. Thus, we obtain $C$ using Formula (2):

$$\prod_{i=1}^{m} (1 + \lambda W_{ij}C) - 1 - \lambda = 0 \ (j = 1, \ldots, n) \tag{2}$$

## 4. Materials and Methods

The questionnaires consist of three parts. The first part contains questions to obtain the respondents' demographic data, such as survey area, status, flight time, nationality, gender, age, and purpose of being a pilot. The second part has the pairwise comparisons made between elements at each level. Pairwise comparisons consist of matrices where first is the main criteria, and then the sub-criteria. Pairwise comparisons were obtained by using the relative importance scale. The respondents were informed about the questionnaire before they answered the questions. The third part of the questionnaire is used to acquire the data for the absolute importance by using the scale from 1 to 7.

The factors for the pairwise comparison of AHP's relative importance that were calculated are composed of categories and sub-categories. SRM as a meta-category has six categories—aeronautical decision making (ADM), risk management (RM), task management (TM), situational awareness (SA), controlled flight into terrain (CFIT) awareness, and automation management (AM). Each category has its own variables (see Figure 2). The ADM category contains the three P variables—perceive, process, and perform. The RM category includes the PAVE variables—the pilot, aircraft, environment, and external pressure.

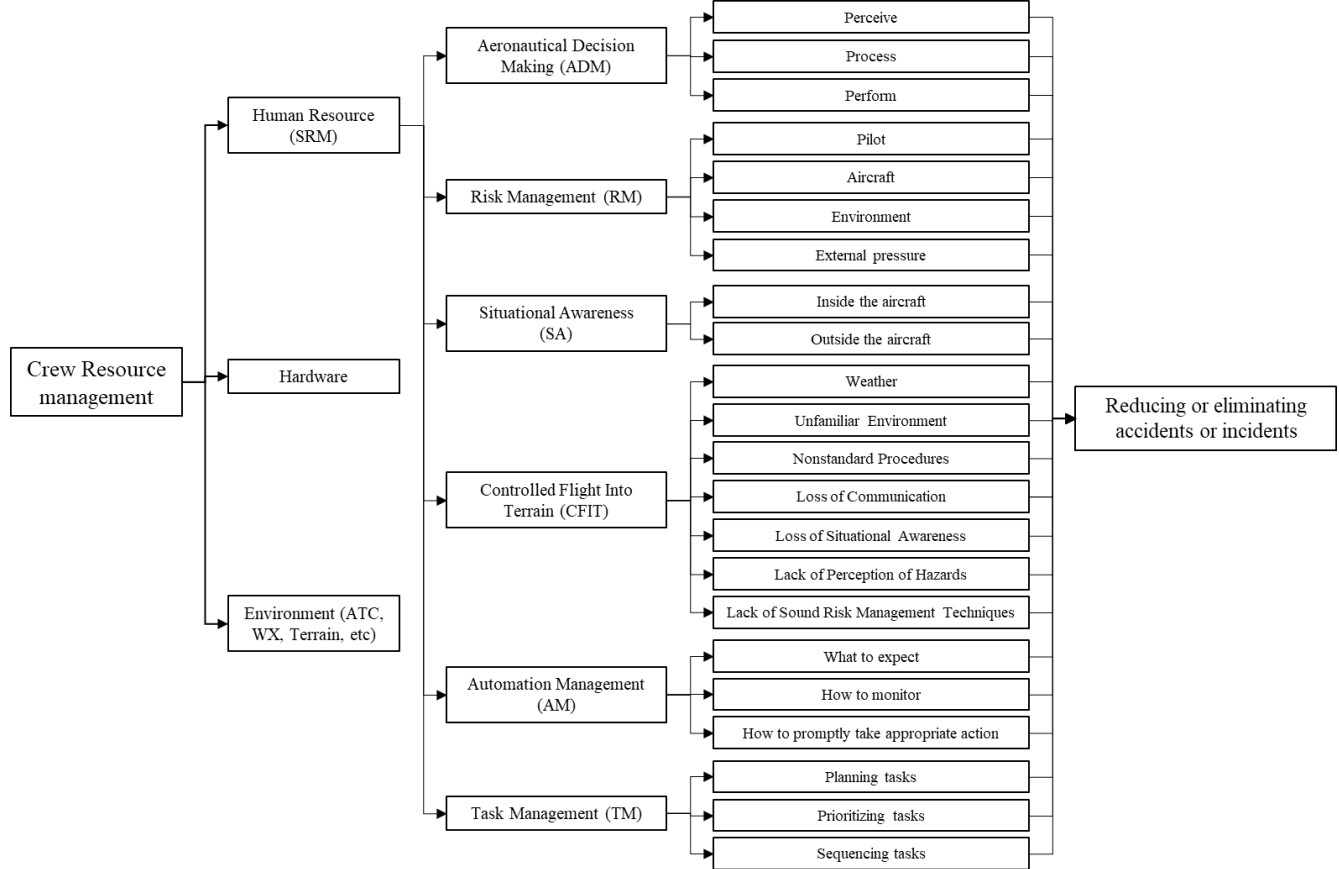

**Figure 2.** Research Model: CRM and SRM's Hierarchical Structure.

The situational awareness category has two variables regarding maintaining the pilot's situational awareness inside the aircraft (the status of the aircraft systems, the pilot, and passengers) and outside the aircraft (awareness of where the aircraft is in relation to the terrain, traffic, weather, and airspace). CFIT, which is attributed to a majority of CFIT accidents, is measured as 1—weather, 2—unknown environment, 3—abnormal procedures, 4—loss or breakdown of communication, 5—loss of situational awareness, 6—absence of perception of hazards, and 7—absence of sound risk management techniques [60]. The automation management category that is used in the ADM model includes (1) what to expect, relating to the peculiarities of the particular automated system being used, (2) how to monitor for proper operations, and (3) how to take appropriate action if the system does not function as expected [19,20,29]. Task management, which means effective workload management to ensure essential operations, is used in the ADM model and has three factors: (1) planning tasks, (2) prioritizing tasks, and (3) sequencing tasks [19,20].

This research survey was performed from 1 August 2018 to 30 July 2019, both in Korea and in the United States. We distributed 400 questionnaires in Korea and the U.S., received 173 responses (43.3%), and considered a valid sample 162 (40.5%). Approximately 63.6% of valid respondents are from Korean pilot training institutions, and 36.4% are from American pilot training institutions located in: Stockton, California; Phoenix, Arizona; and Las Vegas, Nevada. Among them, 48.8% are flight instructors, 19.8% are instrument rating holders, 16% are airline transport pilot license holders (ATP), and 15.4% are commercial rating holders (see Table 1).

**Table 1.** Samples of Respondents.

| Country | Area | Current Status of Respondents | | | | Total |
|---|---|---|---|---|---|---|
| | | Flight Instructor | Commercial Rating | Instrument Rating | ATP | |
| Korea (103, 63.6%) | Hanseo Univ. | 21 | 2 | 8 | - | 31 |
| | Chodang Univ. | 8 | 19 | 4 | - | 31 |
| | Cheonju Univ. | 9 | - | 6 | 2 | 17 |
| | ATP, Pusan | - | - | - | 24 | 24 |
| U.S.A. (59, 36.4%) | Stockton, CA | 10 | 4 | 14 | - | 28 |
| | Phoenix, AZ | 20 | - | - | - | 20 |
| | Las Vegas, NV | 11 | - | - | - | 11 |
| Total | | 79 (48.8%) | 25 (15.%) | 32 (19.8%) | 26 (16.0%) | 162 (100%) |

Most of the respondents (95.1%) are men, while women are only 4.9%. The majority of the respondents' nationality is Korean (90.1%), and 9.9% are American. Approximately 31.9% of the respondents have flight times of less than 250 h, 16.7% have between 251 and 500 h, 24.7% have between 501 and 1000 h, 11.1% have between 1001 and 2000 h, and 16% have between 2001 and 28,000 h. The percentage of respondents between 21 and 25 years old is 21.6%, between 26 and 29 years old is 23.5%, between 30 and 35 years old is 28.4%, between 36 and 40 years old is 10.5%, and between 41 and 63 years old is 16.0%.

## 5. Results of the Hypotheses Analysis

Based on Sections 2 and 3, we utilized six categories of SRM and 22 variables for six categories that influence the pilot's behavior. To measure the relative and absolute importance through pairwise comparisons among categories and variables, AHP and fuzzy AHP techniques were employed. Based on the AHP and fuzzy AHP, we applied an integral fuzzy AHP (Sugeno integral) for the fuzzy measure to obtain a more objective ranking for evaluating the imprecise and vague situations [56,61]. We gathered expert opinions from flight instructors and students in the flight academies in Korea (103 respondents) and the U.S. (59 respondents). We calculated the relative importance of applying pairwise comparisons by the respondents in the context of SRM, which constitutes the evaluation items of ADM, RM, SA, CFIT, AM, and TM. According to [38], the consistency ratio of participants' responses should be considered in the pairwise comparison, which is usually regarded as a reliable response when the consistency ratio is within 0.10.

In this current study, the respondents' consistency ratio was examined, and we calculated a consistency ratio of 0.03, which could be used for the weighted analysis of each category. To calculate the weights, a normalization process was performed to adjust the sum of all weights to one. This allowed us to identify the relative size of individual items within the same hierarchy. The categories were weighted in the same way for both categories and variables. Considering the categories and variables, comprehensive priority was given to the 22 final evaluation variables for SRM (see Table 2).

**Table 2.** Relative Importance of SRM Category and Variables using AHP.

| Category | Relative Importance | | Variables | Relative Importance | | Consistency Ratio | |
|---|---|---|---|---|---|---|---|
| | Korea | U.S. | | Korea | U.S. | Korea | U.S. |
| ADM | 0.183 | 0.273 | Perceive | 0.430 | 0.549 | 0.000 | 0.004 |
| | | | Process | 0.271 | 0.247 | | |
| | | | Perform | 0.299 | 0.203 | | |
| RM (PAVE [1]) | 0.211 | 0.180 | Pilot | 0.450 | 0.495 | 0.002 | 0.005 |
| | | | Aircraft | 0.273 | 0.259 | | |
| | | | Environment | 0.175 | 0.157 | | |
| | | | External pressure | 0.103 | 0.089 | | |
| SA | 0.279 | 0.315 | Inside the aircraft | 0.630 | 0.517 | 0.000 | 0.000 |
| | | | Outside the aircraft | 0.370 | 0.483 | | |
| CFIT | 0.180 | 0.089 | Weather | 0.122 | 0.162 | 0.003 | 0.002 |
| | | | Unknown environment | 0.119 | 0.129 | | |
| | | | Abnormal procedures | 0.108 | 0.083 | | |
| | | | Loss or breakdown of communication | 0.103 | 0.063 | | |
| | | | Loss of situational awareness | 0.281 | 0.323 | | |
| | | | Absence of perception of hazards | 0.181 | 0.163 | | |
| | | | Absence of risk management techniques | 0.086 | 0.077 | | |
| AM | 0.052 | 0.046 | What to expect | 0.345 | 0.405 | 0.000 | 0.003 |
| | | | How to monitor | 0.299 | 0.304 | | |
| | | | How to promptly take action | 0.356 | 0.291 | | |
| TM | 0.095 | 0.096 | Planning tasks | 0.448 | 0.490 | 0.000 | 0.002 |
| | | | Prioritizing tasks | 0.375 | 0.344 | | |
| | | | Sequencing tasks | 0.178 | 0.165 | | |
| CR [2] | 0.002 | 0.006 | - | - | - | - | - |

Note: [1] PAVE (pilot, aircraft, environment, and external Pressure); [2] consistency ratio (CR): combined overall.

To calculate the absolute importance of the fuzzy scale from 7 to 1, we applied a measuring scale as shown in Table 3. The mean value for absolute importance was calculated using an assessment scale (1 to 7) of the questionnaire, which was marked up by each respondent. The absolute importance analysis for SRM categories and variables using the fuzzy scale with Table 3 is calculated and presented in Table 4.

**Table 3.** Measuring Scale for Modifying Relative Importance of AHP.

| Score Point | Scale | Remarks |
|---|---|---|
| 7 | 7: 0.90 6: 0.75 5: 0.60 4: 0.50 3: 0.40 2: 0.30 1: 0.15 | CFIT |
| 6 | 6: 0.90 5: 0.75 4: 0.60 3: 0.45 2: 0.30 1: 0.15 | SRM |
| 5 | 5: 0.90 4: 0.70 3: 0.50 2: 0.30 1: 0.10 | - |
| 4 | 4: 0.90 3: 0.70 2: 0.50 1: 0.30 | RM |
| 3 | 3: 0.90 2: 0.60 1: 0.30 | ADM, AM, TM |
| 2 | 2: 0.9 1: 0.45 | SA |

**Table 4.** Relative Importance of SRM Category and Variables Using Fuzzy AHP.

| Category | Relative Importance | | Variables | Relative Importance | |
|---|---|---|---|---|---|
| | **Korea** | **U.S.** | | **Korea** | **U.S.** |
| ADM | 0.571 | 0.638 | Perceive | 0.762 | 0.768 |
| | | | Process | 0.571 | 0.595 |
| | | | Perform | 0.527 | 0.437 |
| RM (PAVE) | 0.675 | 0.662 | Pilot | 0.832 | 0.853 |
| | | | Aircraft | 0.651 | 0.646 |
| | | | Environment | 0.529 | 0.561 |
| | | | External pressure | 0.385 | 0.341 |
| SA | 0.548 | 0.557 | Inside the aircraft | 0.734 | 0.694 |
| | | | Outside the aircraft | 0.612 | 0.656 |
| CFIT | 0.476 | 0.453 | Weather | 0.528 | 0.544 |
| | | | Unknown environment | 0.503 | 0.490 |
| | | | Abnormal procedures | 0.415 | 0.485 |
| | | | Loss or breakdown of communication | 0.505 | 0.594 |
| | | | Loss of situational awareness | 0.535 | 0.503 |
| | | | Absence of perception of hazards | 0.528 | 0.508 |
| | | | Absence of risk management techniques | 0.560 | 0.476 |
| AM | 0.368 | 0.249 | What to expect | 0.632 | 0.671 |
| | | | How to monitor | 0.691 | 0.564 |
| | | | How to promptly take action | 0.577 | 0.564 |
| TM | 0.584 | 0.651 | Planning tasks | 0.763 | 0.742 |
| | | | Prioritizing tasks | 0.612 | 0.631 |
| | | | Sequencing tasks | 0.425 | 0.427 |

As noted previously, we applied integral fuzzy AHP using λ and C for the absolute importance of the category and variables. We could obtain λ and C using Formulas (1) and (2). Table 5 shows λ and C, multiplying the relative importance and composite importance of all the variables and ranks of each variable for Korea and the U.S. We applied the following formula to obtain a Korea–U.S. combined ranking. $V_{ijk} = n$ for rank 1, $n − 1$ for rank 2, ..., $n − 21$ for rank 22, where $V_{ijk}$ rank score for each variable for $i = 1, 2$ (1 for Korea, 2 for the U.S.). $j = 1, ..., m$ categories and $k = 1, ..., n$ variables. As for the combined score, we

used Formula (3). For the category score, we use Formula (4), where m refers to categories and *n* refers to variables.

$$V_{jk} = V_{1jk} + V_{2jk} \tag{3}$$

$$\Sigma_{k=1}^{n} V_{ij}/n \; \forall j \; (j = 1, \ldots, m) \tag{4}$$

**Table 5.** Composite Importance of SRM Category and Variables using Integral Fuzzy AHP.

| Category | λ | | C | | Variables | Relative Importance | |
|---|---|---|---|---|---|---|---|
| | Korea | U.S. | Korea | U.S. | | Korea | U.S. |
| ADM | −0.931 | −0.921 | 1.861 | 1.676 | Perceive | 0.157 | 0.245 |
| | | | | | Process | 0.099 | 0.110 |
| | | | | | Perform | 0.109 | 0.091 |
| RM (PAVE) | −0.980 | −0.982 | 2.212 | 2.082 | Pilot | 0.156 | 0.174 |
| | | | | | Aircraft | 0.095 | 0.091 |
| | | | | | Environment | 0.061 | 0.055 |
| | | | | | External pressure | 0.036 | 0.031 |
| SA | −0.770 | −0.769 | 3.968 | 4.026 | Inside the aircraft | 0.989 | 0.517 |
| | | | | | Outside the aircraft | 0.581 | 0.483 |
| CFIT | −0.993 | −0.993 | 2.806 | 2.920 | Weather | 0.039 | 0.048 |
| | | | | | Unknown environment | 0.038 | 0.039 |
| | | | | | Abnormal procedures | 0.034 | 0.025 |
| | | | | | Loss or breakdown of communication | 0.033 | 0.019 |
| | | | | | Loss of situational awareness | 0.089 | 0.096 |
| | | | | | Absence of perception of hazards | 0.057 | 0.049 |
| | | | | | Absence of risk management techniques | 0.027 | 0.023 |
| AM | −0.937 | −0.906 | 1.918 | 1.789 | What to expect | 0.030 | 0.035 |
| | | | | | How to monitor | 0.026 | 0.026 |
| | | | | | How to promptly take action | 0.031 | 0.025 |
| TM | −0.921 | −0.919 | 1.551 | 1.738 | Planning tasks | 0.092 | 0.070 |
| | | | | | Prioritizing tasks | 0.077 | 0.049 |
| | | | | | Sequencing tasks | 0.036 | 0.024 |

We applied the integral fuzzy AHP (Table 5) compilation of relative (AHP, see Table 2) and absolute importance (fuzzy AHP, see Table 4) to produce the comprehensive importance of SRM's categories and variables (see Table 6). Column (1) of Table 6 shows the SRM's categories and variables. Column (2) provides the ranking of the variables for Korea and the U.S., along with the differences. Ten variables out of 22 were not found to be different between Korea and the U.S. Most of the variables (18) show differences within ±3. Therefore, the authors do not consider the safety culture, which affects members' attitudes and behavior concerning an organization's safety performance [62], between Korea and the U.S. to be different. Furthermore, 90.1% of respondents are Korean even at American pilot training schools. Column (3) provides the rank score, combined scores (Korea and the

U.S.) based on the ranks in Column (2) using Formula (3), and the ranks of the variables. Column (4) gives the categories' score and rank using Formula (4).

**Table 6.** Combined (Korea and U.S.) Variables and Category Ranks using Integral Fuzzy AHP.

| (1) | | (2) | | | (3) | | | | (4) | |
|---|---|---|---|---|---|---|---|---|---|---|
| | | **Rank** | | | **Rank Score** | | | | **Category** | |
| **Category** | **Variables** | Korea | U.S. | Diff. [1] | Korea | U.S. | Com. [2] | Rank | Score [3] | Rank |
| ADM ($V_1$) | Perceive ($V_{11}$) | 3 | 3 | 0 | 20 | 20 | 40 | 3 | | |
| | Process ($V_{12}$) | 6 | 5 | +1 | 17 | 18 | 35 | 5 | 36 | 2 |
| | Perform ($V_{13}$) | 5 | 8 | −3 | 18 | 15 | 33 | 6 | | |
| RM ($V_2$) [PAVE] | Pilot ($V_{21}$) | 4 | 4 | 0 | 19 | 19 | 38 | 4 | | |
| | Aircraft ($V_{22}$) | 7 | 7 | 0 | 16 | 16 | 32 | 7 | 27 | 3 |
| | Environment ($V_{23}$) | 11 | 10 | +1 | 12 | 13 | 25 | 10 | | |
| | External pressure ($V_{24}$) | 16 | 16 | 0 | 7 | 7 | 14 | 15 | | |
| SA ($V_3$) | Inside the aircraft ($V_{31}$) | 1 | 1 | 0 | 22 | 22 | 44 | 1 | 43 | 1 |
| | Outside the aircraft ($V_{32}$) | 2 | 2 | 0 | 21 | 21 | 42 | 2 | | |
| CFIT ($V_4$) | Weather ($V_{41}$) | 13 | 13 | 0 | 10 | 10 | 20 | 13 | | |
| | Unknown environment ($V_{42}$) | 14 | 14 | 0 | 9 | 9 | 18 | 14 | | |
| | Abnormal procedures ($V_{43}$) | 17 | 19 | −2 | 6 | 4 | 10 | 17 | | |
| | Loss or breakdown of communication ($V_{44}$) | 18 | 22 | −4 | 5 | 1 | 6 | 20 | 16 | 5 |
| | Loss of situational awareness ($V_{45}$) | 9 | 6 | +3 | 14 | 17 | 31 | 8 | | |
| | Absence of perception of hazards ($V_{46}$) | 12 | 12 | 0 | 11 | 11 | 22 | 12 | | |
| | Absence of risk management techniques ($V_{47}$) | 21 | 21 | 0 | 2 | 2 | 4 | 21 | | |
| AM ($V_5$) | What to expect ($V_{51}$) | 20 | 15 | +5 | 3 | 8 | 11 | 16 | | |
| | How to monitor ($V_{52}$) | 22 | 17 | +5 | 1 | 6 | 7 | 19 | 9 | 6 |
| | How to promptly take action ($V_{53}$) | 19 | 18 | +1 | 4 | 5 | 9 | 18 | | |
| TM ($V_6$) | Planning tasks ($V_{61}$) | 8 | 9 | −1 | 15 | 14 | 29 | 9 | | |
| | Prioritizing tasks ($V_{62}$) | 10 | 11 | −1 | 13 | 12 | 25 | 10 | 22 | 4 |
| | Sequencing tasks ($V_{63}$) | 15 | 20 | −5 | 8 | 3 | 11 | 16 | | |

[1] Diff. (difference): (rank of Korea–rank of the U.S.); [2] Com. (combined score) = rank score of Korea + rank score of U.S.; [3] $\Sigma_{k=1}^{n} V_{jk}/n$ for $\forall j$ ($j = 1, \ldots, m$), where $m$ is categories and $n$ is variables.

Based on the data presented in Table 6, the most crucial variable is "Inside the aircraft". The next important variable is "Outside the aircraft". The results show that situational awareness is the most important category that reflects the circumstances of aviation safety in the real world where SRM is applied [9,10,20]. A pilot is expected to examine each situation considering their level of experience, personal minimums, readiness level in terms of current physical and mental conditions, and make their own decision [29]. Poor decision making is the root cause of many—if not most—aviation accidents [29].

On the other hand, good decision making is about avoiding the circumstances that lead to really tough choices. Most pilots have made similar mistakes despite the advanced avionics of their aircraft, which could increase safety with enhanced situational awareness. The errors were prevented before a mishap due to extra margins, sound warning systems, a sharp co-pilot, or just good luck [20]. The single pilot should develop and use situational awareness to avoid information overload.

Aeronautical decision making (ADM) is the next important category with its three P variables—perceive, process, and perform. ADM is an organized framework for risk assessment [19], aiding the decision-making process to improve flight safety [20]. Among the three Ps, the order of importance is as follows: perceive, process, and perform. Using the three Ps, a pilot continuously evaluates every aeronautical decision to recognize and minimize potential threats [10].

Risk management (RM) includes the PAVE variables—pilot, aircraft, environment, and external pressure [10]. The three Ps of ADM and PAVE of RM could be combined [10] to enhance situational awareness [20], and could also be integrated with CARE (consequences, alternatives, reality, and external pressure) and the TEAM (transfer, eliminate, accept, and mitigate) checklist [10]. CARE consists of reviewing hazards and evaluating risks. TEAM involves choosing and implementing controls [10].

Task management (TM) is a crucial component of in-flight safety, where pilots manage multiple tasks that must be carried out safely and efficiently [20]. We find that among the variables of TM, the order of importance is (1) planning, (2) prioritizing, and (3) sequencing tasks. Task management is all about prioritizing and identifying tasks that can be completed before, during, and after a flight to ensure efficient operations without task overload [62].

Controlled flight into terrain (CFIT) is where aircrafts are flown into terrains in a controlled manner, regardless of the crew's situational awareness [32]. Three accident categories account for more than 60% of worldwide fatalities [32,33]. Of these three categories, CFIT is identified as being responsible for nearly one-quarter of all worldwide fatalities, despite representing only 3% of the number of accidents [32,63]. Among the CFIT variables, the respondents perceive that the loss of situational awareness is the most important. The FAA's [63] recommendation for CFIT avoidance is to maintain situational awareness using the five Ps before leaving cruising altitude.

Automation management (AM) is vital for a pilot to use avionics effectively [64]. Automation is an essential advancement in aviation technologies [10]. More pilots now rely on automated flight planning tools rather than traditional flights [10]. Our survey findings show that it is up to the pilot to clarify the expectation of the advanced automation system and maintain proficiency in all tasks.

## 6. Conclusions

This research consists of interviews with student pilots and flight instructors in Korea and the U.S. (Stockton, California; Las Vegas, Nevada; and Phoenix, Arizona). About 400 pilots were selected, and data were collected from questionnaires. Among them, 162 cases were used for analysis for this study. The AHP, fuzzy AHP, and integral fuzzy AHP were applied and analyzed. For the research model, six categories and 22 variables were chosen from the literature review and applied in this study. The importance index of SRM's categories and variables was obtained using AHP and fuzzy AHP through a pairwise comparison with fuzzy scale, fuzzy measure, and fuzzy integral, applying $\lambda$ and parameter C.

This current study provides some explanations and provisions for aviation safety through single pilot resource management. Because there are not many previous works about SRM, it is difficult to determine the devices that give sincere aid to find reasons for various accidents and incidents, which resulted in some difficulties for this research. SRM is a form of CRM training in general aviation that could optimize the single pilot's decision-making processes to increase flight safety and improve flight operation efficiency [24]. The findings from this research indicate that SRM can be shared across decision makers in the general aviation industry and its processes should be considered as safety measures or devices for reducing and eliminating accidents and incidents, both on the ground and in the air.

Among the six SRM categories, we find that situational awareness (SA) is the most important category, followed by the aeronautical decision making (ADM). The pilot's situational awareness should begin before the aircraft leaves the ground because the pilot needs to anticipate what will happen in the future and examine risks and contingencies [9,65]. Among the CFIT variables, the loss of situational awareness is considered the most crucial. Situational awareness is how well a pilot (RM) assesses the situation appropriately inside and outside the aircraft (RM) and achieves safe and efficient flight safety. Although we attempted to find the importance of SRM's categories and variables to see how training pilots and instructors perceive it, all of the SRM-related categories and variables are significant to make flying safer. Aeronautical decision making consists of the three P variables—perceive, process, and perform. The findings from this current study show how pilots perceive potential threats, process each situation, and perform tasks with his or her own decision-making process to minimize the threats along with planning and prioritizing tasks.

For future research, the sample could be enlarged within groups working in the various aeronautical fields and conducted within a different cultural context in terms of region, nation, and organization.

**Author Contributions:** Conceptualization, K.H.I. and W.K.; methodology, K.H.I. and S.-J.H.; software, K.H.I.; validation, S.-J.H. and W.K.; formal analysis, K.H.I.; investigation, S.-J.H.; resources, K.H.I. and S.-J.H.; data curation, K.H.I. and S.-J.H.; writing—original draft preparation, K.H.I.; writing—review and editing, S.-J.H.; visualization, S.-J.H.; supervision, W.K.; project administration, W.K. All authors have read and agreed to the published version of the manuscript.

**Funding:** This research received no external funding.

**Institutional Review Board Statement:** The study was conducted according to the guidelines of the Declaration of Helsinki, and approved by the Institutional Review Board of Hanseo University.

**Informed Consent Statement:** All study participants provided informed written consent prior to survey enrollment.

**Data Availability Statement:** Data available on request due to restrictions of privacy.

**Conflicts of Interest:** The authors declare no conflict of interest.

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
