# Peer review of "A Study on Single Pilot Resource Management Using Integral Fuzzy Analytical Hierarchy Process"

_safety, 2021_

Round 1

Reviewer 1 Report

Lines 11-12: "find alternatives for aviation safety" - I don't think it is correct as we are aiming to enhance/improve aviation safety rather than to find alternatives.

Lines 17-19: "The authors find that Integral Fuzzy AHP provides explicit consideration making up the ambiguity of the linguistic response by the AHP and Fuzzy AHP." - this sentence requires a rewrite, in particular, what does  "making up the ambiguity" mean?

Line 210: I think "?1?1+?2?2+?3?3+⋯???? ⁄ n" should be changed to "(?1?1+?2?2+?3?3+⋯????) ⁄ n"

Line 215: I think "CI = ????−n / n−1" should be replaced by "CI = (????−n) / (n−1)"

Lines 311-312: "22 variables of each category" should be changed to "22 variables in total".

Line 329: "evaluation categories" should be "evaluation variables" as 22 is the number of variables not the number of categories (which is 6).

Line 338: The caption of Table 4 indicates "absolute importance ..."; however, both column headings are "relative importance". A possible typo?

Lines 354-356: "Therefore, the authors do not consider the safety culture, which is thought to affect members’ attitudes and behavior concerning an organization’s safety performance [59] between Korea and the U.S." - this sentence has some grammar issues which requires to be fixed. Furthermore, the link between "safety culture" and the 18 variables with no or small (within+/-3) differences in their ranks are not clear. Most importantly, why the organisational safety culture, a rather critical safety measure, is missing within any of the given 6 categories in the first place? Or is it embedded in some of the categories given such as Risk Management which requires to be explicitly stated?

Lines 356-357: "Furthermore, 90.9% of respondents are Korean even at American pilot training schools." - this sentence may be redundant as it has already stated in Line 304, however, the figures are slightly different, that is, 90.1% in Line 304 but 90.9% in Line 356, which one is correct? Furthermore, the link/logic between this percentage (i.e. 90.9 or 90.1) and the "safety culture" in the sentence prior (pointed out above) is not clear.

Line 362: "a more important" may be changed to "the most important"

Lines 360-366: I have a general question with regard to similar variables from different categories. In particular, CFIT also has a variable (Loss of situational awareness) which is overlapping with the SA category or with its two variables (i.e. Inside the aircraft and Outside of the aircraft). How do you handle this issue properly without double counting and with consistency/robust? I have noticed that the "Loss of situational awareness" variable is the most important variable within CFIT category (see Table 5) which is good; however, its absolute position (rank) has been dropped to 8th (see the 3rd column of Table 6). How do you explain/justify the situation where the two situational awareness variables (i.e. Inside the aircraft and Outside of the aircraft) are having the top 1 and 2 rankings, while the "Loss of situational awareness" variable is only ranked 8th in your analysis?

Page 10: to improve the readability, Table 5 may be moved towards the top of Page 10 instead of at the bottom currently.

Page 11: Table 6, CFIT(V4) has 7 variables but all being marked (V41) which should be (V41), (V42), ... , (V47).

Table 6: the 2nd right column: Category Score(2) should be Score(3)

Line 407: the formula for footnote (3) of Table 6 is incorrect. It should be Formula (4) given at the end of Line 346.

Lines 410-411: Words "in the actual flight field" may be deleted

Lines 419-421: "Because there are not many previous works about SRM, it is difficult to determine the devices that give sincere aid to find reasons for various accidents and incidents, which resulted in some difficulties for this research." - This sentence requires a rewrite as its meaning is not clear currently.

Lines 427-428: "We find that situational awareness (SA) and aeronautical decision-making (ADM) is the most important among the six SRM categories." - This sentence may be changed to "Among the six SRM categories, we find the situational awareness (SA) is the most important category followed by the aeronautical decision-making (ADM)."

Lines 433-435: The sentence "Although we attempted to find the importance of SRM’s categories and variables to see how training pilots and instructors perceive it, all of the SRM-related categories and variables are not negligible to make flying safer." may require a rewrite to clearly express what the authors want to express. In particular, what does "not negligible to make flying safer" mean?

Lines 469-471: The web address of Reference 11 is no longer accessible. It should be changed to "Retrieved on November 10, 2021 (https://nbaa.org/aircraft-operations/safety/vlj-training-guidelines/)"

Author Response

Please see the attached file. Many thanks.

Reviewer 2 Report

1) In line 292, figure 2, there are unused branches in AHP hierarchy (hardware and environment) table.  This figure must be given in 2 separate parts. 

2) Authors must explain how to calculate group decision scores.

3) Mathematical explanations for fuzzy AHP and Integral Fuzzy AHP models are poor. Must be improved.

4) Fuzzy AHP and Integral Fuzzy AHP references are poor.

5) A numeric example is needed for fuzzy calculations.

6) Fuzzy calculations and fuzzy operants must be explained in the paper.

Author Response

Please see the attached file. Many thanks

Reviewer 3 Report

The paper describes an interesting approach analyzing factors which are highly relevant in aviation safety. Using fuzzy set theory is a new and promising way to deal with fuzzy assessment citeria. For the interested reader who is unfamiliar with fuzzy set theory this novel idea is partly difficult to unterstand. It may be therefore helpful to take into account the different preconditions of the readers. Here some suggestions:

Headline: Replacing the abbreviation AHP by Analytical Hierarchy Process

Line 67: At this point the meaning of λ and C is without prior knowledge not understandable. A short introduction to fuzzy set theory with the essentials of the theory may be helpful.

Methodolgy::

Table 2: Concerning to the relative importance of the SRM-variables besides the consistency measures it is of interest of how strong the judgements  between the Korean and U.S. participants do correlate.

Table 4: The headline refers to the absolute importance of SRM categories and variables whereas within the table the headlines of the rows are labeled as relative importance. As in table 2 it is of interest of how strong the values are corrleated (Korean vs. U.S. participants).

Table 5: Here also the intercorrelation of the two data sets is of interest. 

Author Response

(The authors gave the same response as above.)

Reviewer 4 Report

Originality/Novelty: 
The Manuscript and the content of the paper are very well prepared. The topic of the paper is focused on actual challenges on the „Single Pilot Resource Management“ approach, especially in General Aviation. The topic and research question offered by the authors of the paper are as follows: how to evaluate the alternatives for risk management adequately and to solve hazard issues that continuously occur in the field between people, such as pilots, air traffic controllers, and mechanics, and the aircraft in the air or on the ground, is interesting, original and well defined. We feel the lack of scientific studies and a database of verified data from aviation practice on the issue. The originality of the paper can be found in the academic research of Single Pilot Resource Management using MCDA and MCDA methods, including the Analytic Hierarchy Process application. The results of the study contribute to the scientific and academic discussion of the potential of SRM in a wider range of enhancing the quality of situational awareness of pilots, which may be reflected in a higher level of flight safety. The results provide an advance in current knowledge

Significance: 
I can state that the topic and results are interpreted appropriately and are important especially for expanding the knowledge base of experience in the field of SRM planning in general aviation. All conclusions are justified and supported by the results of study. The research question is clear and identified. Last but not least, the elaboration of the topic has a positive effect on the sharing of these results of scientific work. The focus of the article corresponds to the aims and topics of the special issue of the "Safety" journal.

Quality of Presentation: 
The paper is written in an appropriate way based on a relevant review of the literature (61 references), current data, research methods, and analysis of gaps presented in this study on the topic of SRM. The data and analyses are presented appropriately. We can accept standards for the presentation of research results.

Scientific Soundness: 
The research methodology and approach, tools, as the mentioned MCDA and MCDA methods, including the Analytic Hierarchy Process application  (based on criteria for evaluating the alternatives, establishes priorities based on pairwise comparisons of the elements yields a set of overall priorities for the hierarchy, and checks the judgments' consistency, etc.), are also relevant. The paper is correctly designed, technically, and scientifically sound (I would choose the research sections as follows in a different order: Introduction, Materials and Methods, Results, Discussion, Conclusions).
Authors utilize six categories of SRM and 22 variables of each category that influence the pilot’s behavior. Data are relevant to draw conclusions.
Key scientific data have been elaborated on the basis of the research survey performed from August 1, 2018, to July 30, 2019, both in Korea and in the United States. We distributed 400 questionnaires in Korea and the U.S., received 173 responses (43.3%), and considered a valid sample 162 (40.5%). All conclusions are substantiated and based on the results of the research methods. The paper will allow another researcher to reproduce the results (Figure 2. Research Model: Hierarchical Structure, etc.).

Interest to the Readers: 
I can state that the research study and the conclusions are interesting for the readership of the special issue of the „Safety“ journal and attract a wide readership.

Overall Merit: 
From my own experience as flight instructors on both aircraft and helicopters, I can confirm that the results of the study correctly show that situational awareness is a more important category that reflects the circumstances of aviation safety in real practice where SRM is used.
Most general aviation pilots have made and are making similar mistakes despite the aircraft's advanced avionics equipment, which should increase safety with increased situational awareness.
With regard to the opinions of the authors, the present study opens new questions and suggestions for further research into the issue of conditions in general aviation, with an emphasis on aviation schools. I see the article as a contribution to this path of knowledge. The overall benefit of publishing this work is based on an insufficient database of data from practice and an innovative view of the issue, which can be a methodological and praxeological motive, especially for decision-makers, managers of aviation schools, as well as the pilots of general aviation. The study contributes to the scientific and academic discussion on this topic and opens up space for expanding the database of data from the following research, primarily from the environment of flight instructors - pilots (cadets). The study provides an advance towards the current knowledge. The authors have addressed an important actual question of aviation safety with the research survey.
I recommend the paper for publication within the academic and research discussion on the topic.

Author Response

Thank you for the constructive suggestions and comments.

Round 2

Reviewer 1 Report

Small changes required, see attached.

Reviewer 2 Report

I believe the manuscript has been sufficiently improved to warrant publication in Safety.

Reviewer 3 Report

As mentioned in my first review the mathematical background of fuzzy set theory is not easy to understand for those readers who are not familiar with this issue. Perhaps the authors can add a reference like "introduction to fuzzy set theory" or something like that. 

The paper is ready for publication.
